METHODS AND RESOURCES

# Biofilm dispersal patterns revealed using far-red fluorogenic probes

**Jojo A. Prentice** [ID] [1], **Sandhya Kasivisweswaran** [ID] [1], **Robert van de Weerd** [ID] [1,2,3], **Andrew A. Bridges** [ID] [1] *

1 Department of Biological Sciences, Carnegie Mellon University, Pittsburgh, Pennsylvania, United States of America, 2 Ray and Stephanie Lane Computational Biology Department, Carnegie Mellon University, Pittsburgh, Pennsylvania, United States of America, 3 Neuroscience Institute, Carnegie Mellon University, Pittsburgh, Pennsylvania, United States of America

* bridges@cmu.edu

**Data Availability Statement:** The source data used to generate all main and supporting figures in this work are available on Figshare (https://figshare.com/s/e0978ade2bc95dccf357). Biological

## Abstract

Bacteria frequently colonize niches by forming multicellular communities called biofilms. To explore new territories, cells exit biofilms through an active process called dispersal. Biofilm dispersal is essential for bacteria to spread between infection sites, yet how the process is executed at the single-cell level remains mysterious due to the limitations of traditional fluorescent proteins, which lose functionality in large, oxygen-deprived biofilms. To overcome this challenge, we developed a cell-labeling strategy utilizing fluorogen-activating proteins (FAPs) and cognate far-red dyes, which remain functional throughout biofilm development, enabling long-term imaging. Using this approach, we characterize dispersal at unprecedented resolution for the global pathogen *Vibrio cholerae*. We reveal that dispersal initiates at the biofilm periphery and approximately 25% of cells never disperse. We define novel micro-scale patterns that occur during dispersal, including biofilm compression during cell departure and regional heterogeneity in cell motions. These patterns are attenuated in mutants that reduce overall dispersal or that increase dispersal at the cost of homogenizing local mechanical properties. Collectively, our findings provide fundamental insights into the mechanisms of biofilm dispersal, advancing our understanding of how pathogens disseminate. Moreover, we demonstrate the broad applicability of FAPs as a powerful tool for high-resolution studies of microbial dynamics in complex environments.

## Introduction

Biofilms are surface-attached, matrix-encapsulated communities of bacteria that are ubiquitous in the environment and are notorious for causing infections [1,2]. Existing in the biofilm state allows encapsulated cells to function as a unit, collectively acquiring nutrients and resisting external threats [3–6]. These emergent, group-level properties make biofilm-forming bacteria difficult to eradicate in clinical and industrial settings [7,8]. The biofilm lifecycle is a dynamic process that occurs in developmental stages: founder cell attachment, biofilm growth, and dispersal [9]. Biofilm dispersal remains under-investigated but is crucial for the

materials used in this study are available upon request from Carnegie Mellon University. All custom processing scripts are available on Github (https://github.com/BridgesLabCMU/Biofilm-dispersal) or Zenodo (https://zenodo.org/records/14012186).

**Funding:** This work was supported by NIH grant R00AI158939 to AAB, a Shurl and Kay Curci Foundation grant to AAB (https://curcifoundation.org/), a Kaufman Foundation New Investigator Research Grant to AAB KA2023-136488 (https://kaufman.pittsburghfoundation.org/), a Damon Runyon Cancer Research Foundation Dale F. Frey Award for Breakthrough Scientists to AAB 2302-17 (https://www.damonrunyon.org/), and startup funds from Carnegie Mellon University to AAB. The funders had no role in study design, data collection and analysis, decision to publish, or preparation of the manuscript.

**Competing interests:** I have read the journal's policy and the authors of this manuscript have the following competing interests: RVDW is a co-founder of Biocognon, Inc.

**Abbreviations:** FAP, fluorogen-activating protein; LB, lysogeny broth; MGe, malachite green ester; MG-2P, malachite green 2 PEG; scFV, single-chain fragment variable; ssMBP, secretion signal of maltose-binding protein.

dissemination of bacteria to new niches and new hosts [10]. Recently, high-throughput screening approaches were used to identify the genetic components that orchestrate biofilm dispersal in the global pathogen and model organism *Vibrio cholerae* [11]. Thus, we now know several molecular players involved in the biofilm dispersal decision, yet how this dynamic process is patterned and executed in space and time remains mysterious.

Previous work has characterized single-cell trajectories and gene expression dynamics in biofilms using fluorescent protein fusions, a technology that exhibits numerous limitations for long-term imaging of biofilm development [12,13]. One major obstacle is that traditional fluorescent proteins are sensitive to environmental changes that occur during biofilm growth, such as oxygen limitation, leading to diminishing fluorescence signal as the community develops [14,15]. However, other, less commonly utilized fluorescent tools, do not exhibit such limitations. One example, which we leverage in the present work, is fluorogen-activating proteins (FAPs). FAPs are engineered single-chain fragment variable (scFV) antibodies that bind to and activate the fluorescence of otherwise non-fluorescent small molecule fluorogens supplied in growth media (Fig 1A) [16–18]. In contrast to traditional fluorescent proteins, FAPs do not require chromophore maturation and are therefore unlikely to be affected by oxygen deprivation, complexes are brighter and more photostable than fluorescent proteins, and finally, numerous FAP-fluorogen complexes are available that span the visible spectrum [19,20]. Far-red probes are particularly useful, as microbial growth medias exhibit low background in these wavelengths, red excitation light is less toxic to growing cells compared to shorter wavelength light, and this region of the visible spectrum enables imaging at greater depths [21].

Here, we demonstrate the utility of far-red FAP-fluorogen complexes for probing biofilm dynamics at high spatiotemporal resolution. To this end, we introduced a constitutively expressed FAP sequence into the genome of *V. cholerae*. Growth of strains expressing the FAP in the presence of malachite green-derived fluorogens yielded intense and stable far-red fluorescence from cells, which enabled us, for the first time, to examine large biofilm communities at single-cell resolution throughout dispersal. We find that biofilm dispersal is heterogeneous in *V. cholerae*; by its completion, approximately 75% of cells have exited the biofilm matrix. The remaining cells are enriched in the biofilm core due to cell departure from the biofilm periphery early in dispersal. We show that heterogeneity in cell motions and overall biofilm compression are fundamental features of dispersal in *V. cholerae*, and we demonstrate that these patterns are attenuated in biofilm dispersal mutants with disparate molecular functions. Together, this work demonstrates the utility of FAP technology in studying microbial community development and reveals how the collective process of *V. cholerae* dispersal is executed at the scale of single cells.

## Results

### FAP-based cell labeling enables long-term imaging of bacterial biofilm development

To overcome the limitations of traditional fluorescent proteins, we explored the utility of a far-red FAP-fluorogen complex for constitutive bacterial cell labeling. We utilized the dL5 FAP and its cognate cell-permeable fluorogen malachite green ester (MGe), which exhibits excitation and emission peaks of 636 nm and 664 nm, respectively, when bound to dL5 [16,22]. For initial comparison to a representative fluorescent protein, we chromosomally integrated *dL5* as a fusion to *mNeonGreen* driven by the tac promoter in *V. cholerae*. When this strain was grown in the presence of norspermidine, which drives biofilm formation [23,24], we observed that mNeonGreen signal in biofilms rapidly decreased after approximately 9 h of growth (Fig 1B). By contrast, the same biofilms exhibited stable far-red fluorescence that increased in-

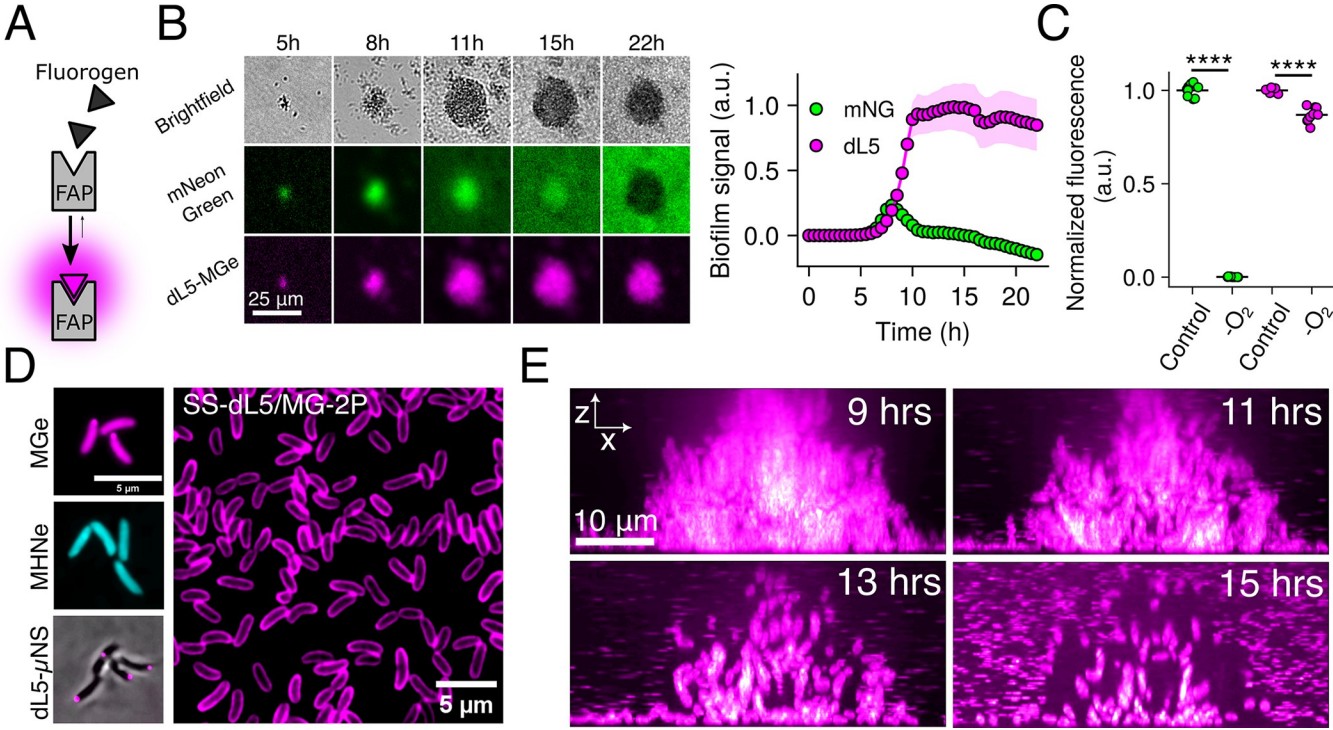

**Fig 1. FAPs facilitate long-term imaging of bacterial communities.** (A) Schematic for FAP-fluorogen interaction and initiation of fluorescence signal. (B) Left panel: Images of *V. cholerae* biofilm growth at low magnification (10× objective) for a strain constitutively expressing a representative fluorescent protein, mNeonGreen, fused to the FAP dL5, grown in the presence of 1 μM MGe and 100 μM norspermidine to stimulate biofilm formation. Right panel: quantification of whole-biofilm fluorescence intensity for the same images. Points represent averages and shaded regions represent standard deviations. $N = 4$ biological replicates. (C) Final time point bulk culture fluorescence for the same strain as in B grown for 16 h in the presence or absence of oxygen. Fluorescence for each channel is normalized to oxic conditions. $N = 3$ biological, 3 technical replicates. $P = 2.5 \times 10^{-13}$ and $4.0 \times 10^{-6}$ for mNeonGreen and dL5-MGe, respectively, based on two-sided unpaired *t* tests relative to the oxic controls. (D) Left panels: Representative images of cells expressing cytoplasmic dL5, labeled with 1 μM MGe (top), 5 μM MHNe (middle), or fused to μNS and labeled with 1 μM MGe (bottom). Right panel: Field of view of cells expressing SS-dL5 labeled with 1 μM MG-2P. (E) Side-on (x-z) view of high-resolution confocal micrographs of *V. cholerae* biofilm cells expressing SS-dL5 labeled with 1 μM MG-2P over the course of biofilm dispersal. The entire biofilm was optically sectioned for 16 h with 10-min intervals. Images are displayed with a magenta-hot lookup table. a.u., arbitrary units. ****$P < 0.0001$. Underlying data for this figure can be found on Figshare (https://figshare.com/s/e0978ade2bc95dccf357). FAP, fluorogen-activating protein; MGe, malachite green ester.

step with increasing biofilm biomass until stationary phase was reached at approximately 16 h postinoculation (Fig 1B). Thus, the dL5-MGe labeling approach is not susceptible to the environmental conditions that lead to mNeonGreen dimming, which likely occurs due to oxygen depletion in dense bacterial communities [25]. Consistent with this observation, we found that growth of this strain in hypoxic conditions eliminated mNeonGreen signal, whereas the dL5-MGe complex remained fluorescent (Fig 1C). Together, these results indicate that the dL5 FAP, in conjunction with its fluorogen MGe, can be used to measure long-term biofilm dynamics, unlike traditional fluorescent proteins. Of note, we found that expression of dL5 in *Escherichia coli* and *Pseudomonas aeruginosa* and subsequent growth in the presence of MGe yielded fluorescent cells, demonstrating the utility of this labeling strategy in other important biofilm formers (S1A and S1B Fig).

The dL5 FAP has previously been shown to bind to additional fluorogens, each of which exhibits unique characteristics. Among these, MHN-ester (MHNe), a cell-permeable fluorogen, exhibits green fluorescence upon dL5 binding ($\lambda_{ex} = 456$ nm, $\lambda_{em} = 532$ nm) [26]. We found that the addition of MHNe to a *V. cholerae* strain carrying *mNeonGreen*[Y69G]-*dL5*, which harbors an inactivating point-mutation in the chromophore region of mNeonGreen, resulted

in cells that exhibited green fluorescence, demonstrating that the same FAP (dL5) can be labeled with spectrally distinct fluorochromes (Fig 1D) [27]. This feature provides flexibility when labeling with multiple probes. To explore whether dL5 is functional when targeted to specific subcellular localizations, we fused dL5 to μNS, an avian virus protein, which, when expressed in bacteria, results in punctate cytoplasmic localization [28]. This approach has previously been used via fusion to fluorescent proteins to monitor cytoplasmic diffusion and cell positions in biofilms [13,28]. As expected, the addition of MGe to this strain yielded bright foci within the cytoplasm (Fig 1D). Finally, we targeted dL5 to the periplasm by generating a fusion to the secretion signal of the *V. cholerae* maltose-binding protein, which we refer to as SS-dL5. Of note, many fluorescent proteins do not fluoresce in the periplasm due to conditions hostile to chromophore maturation [29]. Addition of a pegylated, cell-impermeable version of malachite green (MG-2P) to cells harboring SS-dL5 yielded far-red fluorescence labeling at the cell periphery (Fig 1D) [22]. Importantly, none of the labeling approaches described here impacted the growth of *V. cholerae* (S2 Fig and S1 Text). Together, these results demonstrate the spectral and targeting versatility of the dL5 FAP for labeling bacterial cells.

Given the advantages of the FAP-fluorogen labeling approaches described here, we set out to leverage these tools to define the steps of biofilm dispersal with high spatiotemporal resolution. We chose to utilize the periplasmic labeling approach described above (SS-dL5/MG-2P) to define cell positions in biofilms, as this strategy yielded the highest cellular fluorescence intensity and fastest biofilm labeling kinetics of the approaches explored here (S3 Fig and S1 Text). Using spinning-disc confocal microscopy, we followed the development and dispersal of unperturbed *V. cholerae* biofilms over a 24-h period with continuous volumetric imaging at 10-min time intervals, roughly 2× faster than cells divide in exponential phase (Fig 1E and S1 Movie). Consistent with previous reports, we found that under the static growth conditions used here, *V. cholerae* formed biofilms early in culture growth, with biomass peaking at 9 h post-inoculation. Subsequently, biofilms began to spontaneously disperse due to the accumulation of signals that drive dispersal, and dispersal reached completion at 16 to 17 h postinoculation (Fig 1E) [11,12,30]. For the remainder of this work, we take the time point of peak biofilm biomass to be time point zero of dispersal.

## Dispersal initiates at the biofilm periphery, followed by radially random departure

To define the progression of biofilm dispersal in *V. cholerae*, we first set out to determine the fraction of biofilm biovolume that disperses in the wild-type strain. By quantifying the biofilm biovolume over time based on high-resolution timelapses of 5 wild-type biofilms, we found that approximately 75% of the biovolume dispersed in the wild-type strain (Fig 2A). Thus, in all tested replicates, a subpopulation of cells remained in the biofilm, presumably because of physical entrapment in the remaining matrix and/or the adoption of a distinct physiological state. Regarding the rate of cell departure, we observed variability between biological replicates (Fig 2A). In most cases (3/5), dispersal was initially rapid for the first ~5 h and slowed thereafter. In other cases (2/5), cell departure was roughly constant throughout the dispersal process. Together, these results indicate that the rate of dispersal can vary between biofilms and likely depends on variations in matrix structure, but ultimately, all tested wild-type biofilms retained approximately 25% of cell biovolume at the final time point.

We wondered how dispersal is spatially patterned with respect to distance from the biofilm core. One possibility is that cells in the biofilm core could be the first to disperse (referred to as an "inside-out" scenario), as these cells are more densely packed and are therefore likely the first to experience starvation and cell-density signals that are known to drive dispersal (Fig 2B)

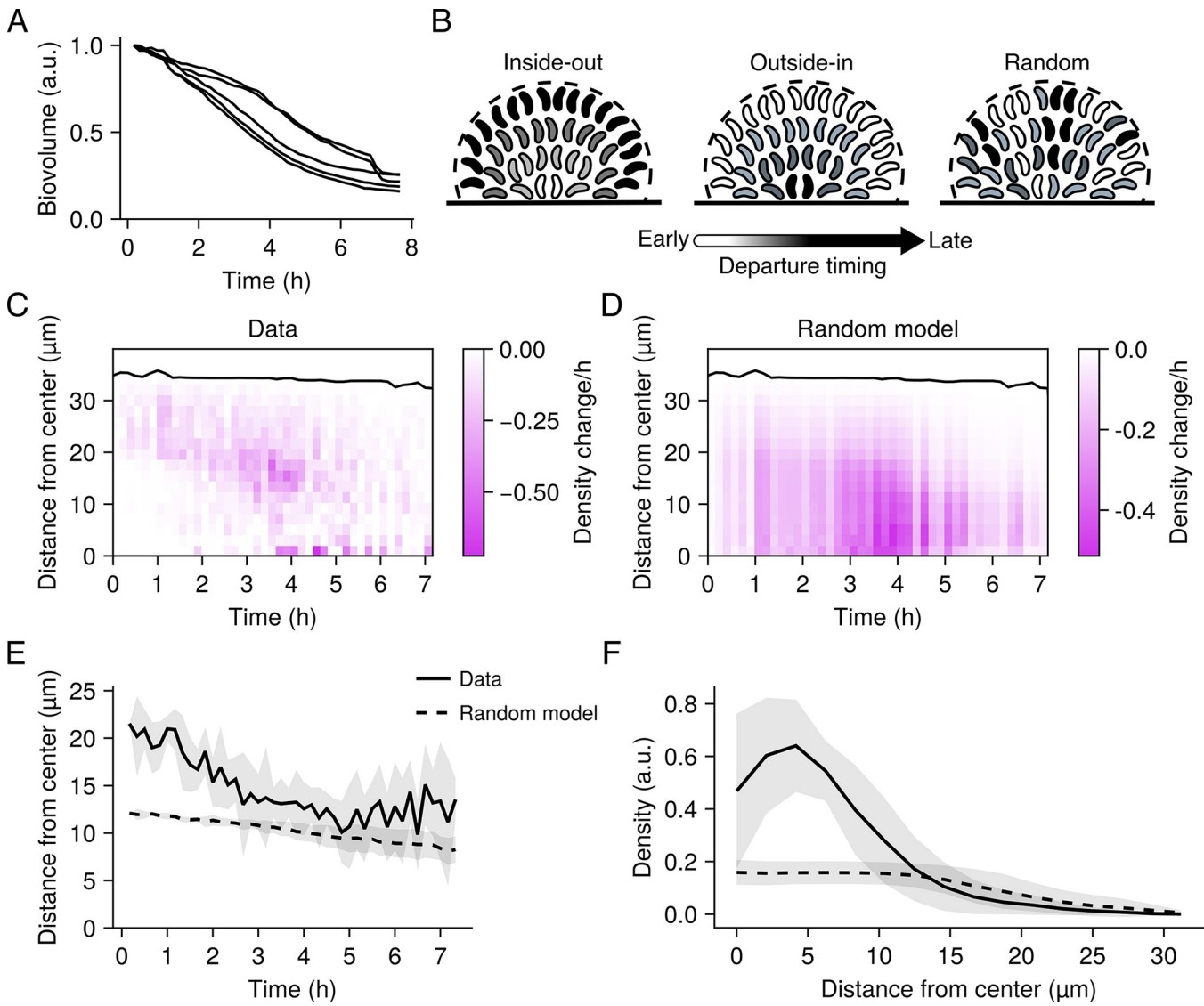

**Fig 2. Radial patterning of wild-type *V. cholerae* dispersal.** (A) Change in total biovolume (cellular volume inside a biofilm) of individual wild-type biofilms over time as measured from spinning-disc confocal sections. Data for each biofilm are normalized as fold-change relative to the biovolume at the initiation of biofilm dispersal. *N* = 5 biological replicates. For all panels pertaining to dispersal, time point zero is the time point of peak biofilm biomass. (B) Schematics of the proposed radial dispersal patterns. Timing of cell departures is indicated by shading, with light regions departing early and dark regions departing late. "Inside-out" corresponds to a scenario in which dispersal progresses from the core cells to the peripheral cells. "Outside-in" corresponds to a scenario in which dispersal progresses from the peripheral cells to the core cells. "Random" corresponds to a scenario in which cells depart the biofilm via a radially random process. (C) Representative kymograph of dispersal from a single wild-type experimental replicate, representing the change in local density at 10-min intervals and at the indicated distances from the biofilm core. Black line represents the biofilm boundary. (D) As in C for the "Random" model. The rate of overall cell departure was set to match the experimental data. (E) Centroid position of the density changes from the experimental data and the model of random cell departures over time. Lines and shading represent means and standard deviations of *N* = 5 biofilms, respectively. (F) Spatial distribution of the local density profiles for the data and random model at the completion of dispersal. Lines and shading represent means and standard deviations for *N* = 5 biofilms, respectively. a.u., arbitrary units. Underlying data for this figure can be found on Figshare (https://figshare.com/s/e0978ade2bc95dccf357).

[12,30]. Alternatively, cells at the biofilm periphery could disperse first (referred to as the "outside-in" scenario), given their proximity to the exterior (Fig 2B). Finally, it is possible that the distance of a cell from the biofilm core does not impinge on its dispersal timing (referred to as a "random" scenario) (Fig 2B). To distinguish between these possibilities, we compared the spatial progression of biofilm dispersal from our timelapse data to a simulation of spatially

random dispersal, which is sufficient to determine whether, at every time point, dispersal is occurring closer to or further from the biofilm core than expected based on chance. Specifically, we computed local changes in the biofilm volume fraction (i.e., the density) relative to the distance from the biofilm center over time and displayed the results as a kymograph (Figs 2C and S4). We used the departure rate calculated from our experimental results to counterfactually simulate the random model (Figs 2D and S4). In our experimental results, cell departure predominated in the biofilm periphery for the first ~3 to 4 h of biofilm dispersal, while less dispersal occurred in the biofilm core than expected from the random model (akin to a modest outside-in trend) (Fig 2C and 2D). After this point, the experimental results resembled the random model. To quantitatively compare the spatiotemporal pattern in the experimental data to the random model, we extracted the centroid of the spatial density changes at each time point from the experimental and model kymographs and evaluated their similarity (Figs 2E and S4). These quantities provide information about where, on average, cells dispersed from in each biofilm and at each time point. For the duration of biofilm dispersal, cell departures occurred closer to the biofilm periphery than suggested by the random model, and we observed a convergence between the data and the random model as dispersal proceeded (Figs 2E and S4). Furthermore, after dispersal reached completion, remaining cells were enriched in the biofilm core relative to the expectation of the random model (Figs 2F and S4). These results suggest that dispersal of peripheral cells is partly a precondition for dispersal of core cells in *V. cholerae* biofilms, in contrast to observations in other bacterial species [31]. However, the process is not akin to a depolymerization mechanism where cells depart in a highly ordered fashion.

## Biofilm compression and regional heterogeneity in cell motions pattern dispersal

Although the results above establish how wild-type *V. cholerae* dispersal is patterned with respect to the core-periphery axis, they provide no insight into non-radial trends. Thus, we next evaluated the prevalence of non-radial patterns in *V. cholerae* biofilm dispersal. We noticed that during the dispersal process, the overall biofilm structure appeared to compress (Fig 3A and S1 Movie), with non-dispersing cells moving towards the biofilm core. We sought to distinguish this apparent inward motion from dispersal of peripheral biofilm cells. By quantifying local displacements in the biofilms using image cross-correlation, we determined that cell motion tended to orient away from the biofilm core during biofilm growth, consistent with cells being pushed outward as the biofilm expands [13]. In contrast, as dispersal initiated, we observed that motions of remaining cells reoriented toward the biofilm core, particularly when we observed dispersal events occurring within the biofilm core (Figs 3B, 3C, and S5). We surmise that cells themselves act as a major source of mechanical support for the biofilm structure, so that as cells regionally depart, neighboring areas compress to fill the open space. In addition, we consistently observed regional heterogeneity in cell displacements during the dispersal process. Specifically, we noticed biofilm regions where outward cell displacements were greater than surrounding areas, presumably reflecting areas that are particularly amenable to dispersal (Fig 3D and S2 Movie). We define these "dynamic regions" as areas where outward displacements were greater than a threshold cutoff. When we quantified the biofilm volume fraction occupied by dynamic regions, we found that their prevalence was highest at the onset of dispersal and steadily decreased as dispersal proceeded, consistent with the slowing of dispersal over time (Fig 3D). We propose 2 mechanisms that could explain the development of dynamic regions: (1) the local properties of the biofilm matrix stochastically vary, leading to regions that are particularly amenable to dispersal; and/or (2) dynamic regions are composed of cell lineages that actively adopt distinct physiological/gene expression states, making them

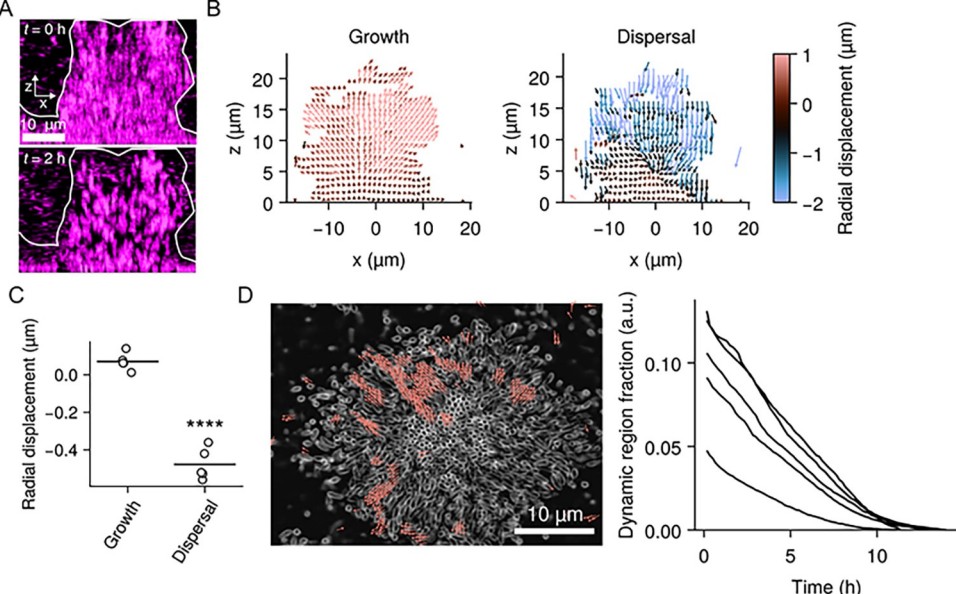

**Fig 3. Compression and regional heterogeneity in cell motions pattern *V. cholerae* biofilm dispersal.** (A) Representative x-z spinning-disc confocal projections from of the first 2 h of dispersal of a wild-type *V. cholerae* biofilm highlighting compression of peripheral cells toward the core. The boundary of the biofilm is highlighted in white. Scale is as indicated. (B) Net displacement vectors during a ~1.5-h period preceding dispersal ("Growth") and throughout dispersal ("Dispersal"). Color represents radial displacement (μm) with respect to the core of the biofilm. For ease of demonstration, a single projection on the x-z plane is shown. (C) Quantification of the radial components of the displacements during the growth and dispersal phases. $N = 5$ replicate biofilms. For each replicate, values are averaged over the entire biofilm. $P = 1.06 \times 10^{-5}$ based on a two-sided unpaired *t* test relative to the "Growth" data. (D) Left panel: representative x-y slice of a spinning-disc confocal z-stack from a timelapse of wild-type *V. cholerae* dispersal, with net displacement vectors from the first 4 h of the timelapse overlayed. A length threshold was applied to isolate locally elevated displacements, taken to be a measure of dynamic regions in biofilms. Scale is as indicated. Right panel: quantification of dynamic region volume fraction during the 40 min preceding and throughout the dispersal phase for wild-type *V. cholerae*. Each line represents an independent replicate for $N = 5$ biofilms. a.u., arbitrary units. ****$P < 0.0001$. Underlying data for this figure can be found on Figshare (https://figshare.com/s/e0978ade2bc95dccf357).

more likely to disperse. Together, the compression and regional variability in cell motions observed during *V. cholerae* biofilm dispersal suggest a model in which, during biofilm development, certain areas of biofilms become more fluid-like, enabling localized outward motion of cells even from the internal regions of the biofilms. At the same time, the more rigid cell groups undergo compression to fill newly unoccupied space. How these localized differences in mechanical properties are established during biofilm development/dispersal will be the subject of future work.

## Biofilm dispersal defect mutants exhibit altered patterns during the dispersal phase

Having defined the heterogeneous patterns underlying wild-type biofilm dispersal, we sought to determine how distinct molecular mechanisms affect specific aspects of the dispersal process. Previously, we and others identified several dispersal defective mutants, three of which we investigate here: Δ*cheY3* (which we refer to as Δ*cheY*), Δ*lapG*, and Δ*rbmB*. *cheY* encodes a response regulator protein that controls chemotaxis and swimming directionality [11]. When phosphorylated, CheY binds to the flagellar motor, driving a transition from counterclockwise to clockwise flagellar rotation (Fig 4A). Thus, in the Δ*cheY* strain, cell swimming is

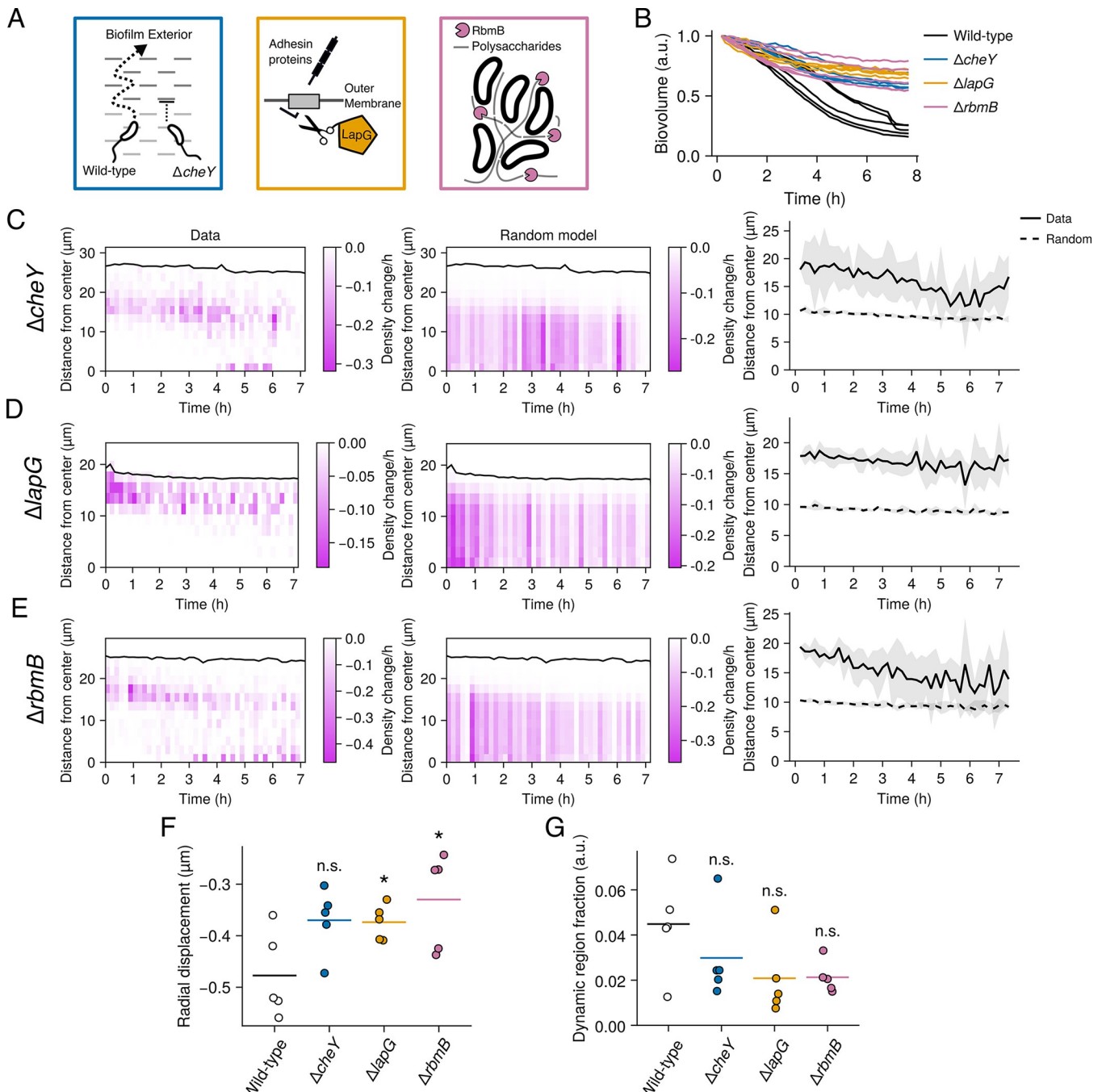

**Fig 4. Dispersal patterns in the Δ*cheY*, Δ*lapG*, and Δ*rbmB* mutants.** (A) Cartoon depiction of the proposed roles of CheY, LapG, and RbmB in *V. cholerae* biofilm dispersal. (B) Change in total biovolume (cellular volume inside a biofilm) of individual biofilms for the indicated strains over time based on spinning-disc confocal z-stacks. The data for each biofilm are normalized as fold-change relative to the biovolume at the first time point. *N* = 5 biological replicates. (C) Left panel: representative kymograph of dispersal data from a Δ*cheY* replicate, depicting the change in local density at 10-min intervals and at the indicated distances from the biofilm core. Black line represents the biofilm boundary. Middle panel: as in the left panel for a model of random cell departures. Right panel: centroids of the density changes from the experimental data and the model of random cell departures over time. Lines and shading represent means and standard deviations for *N* = 5 biofilms, respectively; (D) as in C for the Δ*lapG* mutant; (E) as in C for the Δ*rbmB* mutant. (F) Quantification of the radial components of the displacements during the dispersal phase for the indicated strains. Crossbars represent means and dots represent individual replicates for *N* = 5 biofilms, with the radial displacements averaged over each replicate biofilm. *P* = 0.054, 0.030, and 0.048 based on two-sided unpaired *t* tests of the Δ*cheY*, Δ*lapG*, and Δ*rbmB* data relative to the wild-type data, respectively. (G) Quantification of dynamic region volume fraction during the dispersal phase for the indicated strains. Data represent averages over the entire dispersal phase. Crossbars represent means and dots represent individual replicates for *N* = 5 biofilms. *P* = 0.291, 0.094, and 0.072 based on two-sided unpaired *t* tests of the Δ*cheY*, Δ*lapG*, and Δ*rbmB* data relative to the wild-type data, respectively. a.u., arbitrary units. *\*P* < 0.05; n.s., *P* > 0.05. Underlying data for this figure can be found on Figshare (https://figshare.com/s/e0978ade2bc95dccf357).

unidirectional. Moreover, previous genetic evidence has demonstrated that the dispersal defect of this mutant is not due to its inability to follow a chemotactic gradient, but rather is due to the mutants' inability to reorient its swimming directionality [11]. Thus, it is proposed that frequent reorientations in swimming direction enable cells to escape the biofilm matrix as it is digested. By contrast, the genes *lapG* and *rbmB* encode matrix digestion enzymes. Specifically, *lapG* encodes a conserved periplasmic protease that degrades outer-membrane-spanning adhesins that mediate cell-surface attachments, and possibly also cell–cell connections (Fig 4A) [32–34]. Thus, in the Δ*lapG* strain, adhesins remain intact during dispersal, rendering the process defective. Finally, *rbmB* encodes an extracellular polysaccharide lyase, which is proposed to degrade the polysaccharide component of the *V. cholerae* biofilm matrix (Fig 4A) [35,36]. Thus, in the Δ*rbmB* strain, the polysaccharide component of the biofilm matrix is presumably intact during dispersal. Notably, although the 3 mutants affect biofilm dispersal in *V. cholerae*, they do not alter the peak biofilm size and are therefore ideal candidates for specifically perturbing the dispersal phase of biofilm development [11].

As with our analysis of the wild-type dispersal pattern, we first examined the change in total biofilm volume over time for each of the mutants. All 3 mutants exhibited notable dispersal defects, consistent with previous findings (Fig 4B and S3–S5 Movies) [11]. The magnitude of the dispersal defect in the Δ*cheY* strain was higher than previously reported, with approximately 65% of cells remaining in the biofilms after the dispersal phase, on average—a defect similar to the Δ*lapG* and Δ*rbmB* strains (Fig 4B) [11]. Next, we considered the spatial progression of dispersal in these mutants. An analysis of their radial dispersal patterns revealed that cell departures predominated at the biofilm periphery over the course of the dispersal phase compared to the results of the random model (Figs 4C–4E and S6–S8). The Δ*cheY* and Δ*rbmB* mutants did display modest convergence to the random model as dispersal progressed, akin to the wild-type (see Fig 2E), whereas the Δ*lapG* mutant exhibited cell departures that strictly occurred at the biofilm periphery throughout the dispersal phase (Figs 4C–4E and S6–S8). Together, these results indicate that swimming reorientations, mediated by CheY, and polysaccharide degradation, mediated by RbmB, are required for proper dispersal but do not dramatically alter radial dispersal trends. In contrast, adhesin cleavage by LapG is required for cells located in the biofilm core to exit.

In addition to perturbing the radial patterning of dispersal, a logical prediction is that the dispersal mutants should exhibit attenuated biofilm compression, since biofilm compression presumably depends on the amount of dispersal occurring between the biofilm core and periphery (Fig 3B). Consistent with this prediction, we observed that on average, the radial components of the displacement vectors with respect to the biofilm core during the dispersal phase were more strongly negative in the wild-type strain compared to the dispersal mutants, reflecting attenuated compression in the mutants (Fig 4F and S3–S5 Movies). We surmise from these results that biofilm compression is generally dependent on the amount of cellular dispersal. An additional prediction arising from the results in Fig 3 is that the dynamic regions observed in the wild-type strain should also either be attenuated or decoupled from dispersal in the Δ*cheY*, Δ*lapG*, and Δ*rbmB* mutants. Indeed, the total volume fraction of biofilms occupied by dynamic regions was, on average, lower in the 3 dispersal mutants compared to the wild-type strain, though due to variation between replicates, the comparisons did not attain statistical significance (Fig 4G, $P > 0.05$ in all cases).

## Disruption of cell–cell attachments results in complete dispersal and abrogates patterns

Having shown that the joint activities of proteins with disparate functions are necessary to drive wild-type dispersal patterns, we wondered whether the reverse is true—that is, whether

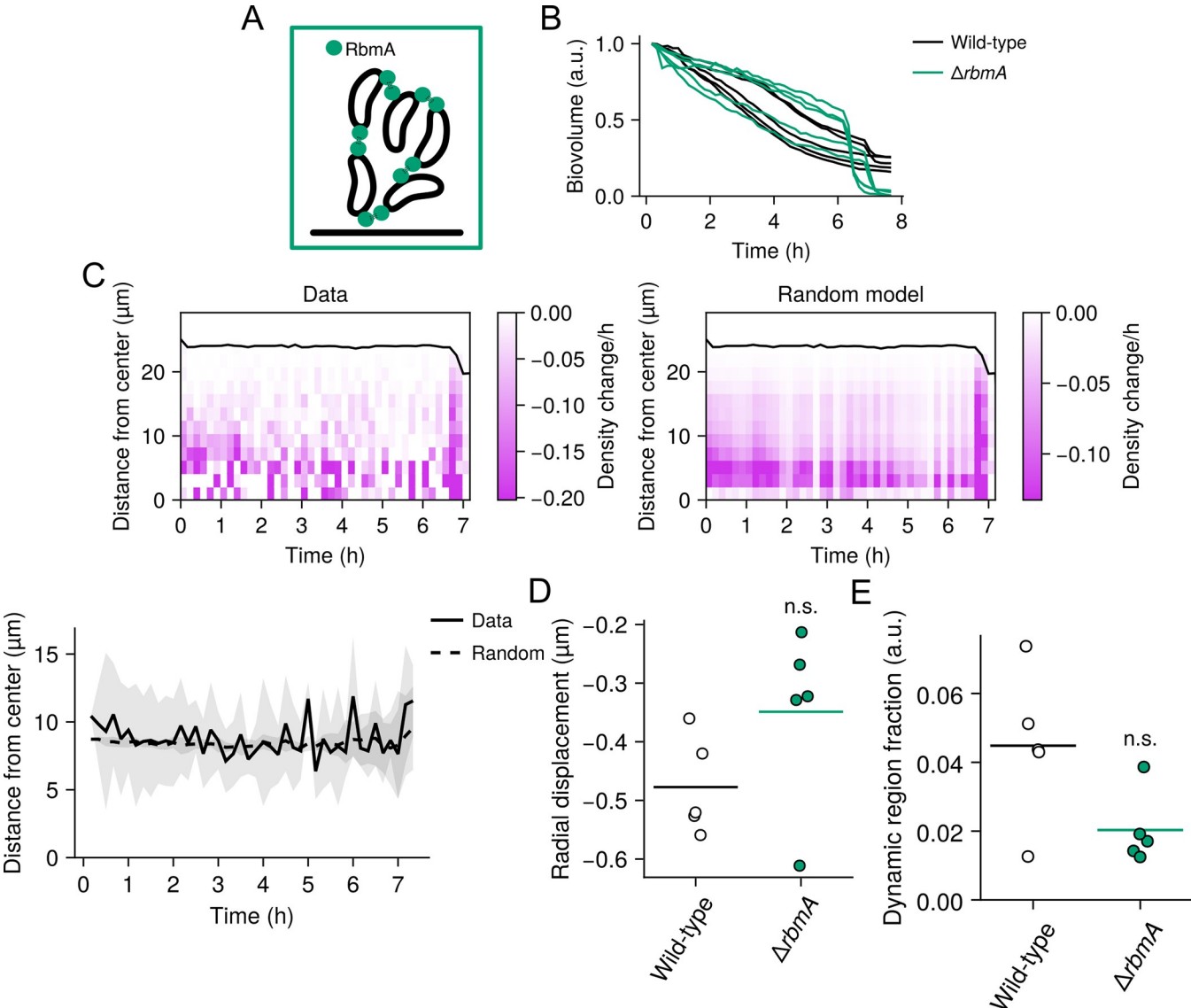

**Fig 5. Dispersal patterns in the Δ*rbmA* mutant.** (A) Cartoon depiction of the role of RbmA in forming cell–cell connections. (B) Change in total biovolume (cellular volume inside a biofilm) of individual biofilms for the indicated strains over time based on spinning-disc confocal z-stacks. The data for each biofilm are normalized as fold-change relative to the biovolume at the first time point. *N* = 5 biological replicates. (C) Left panel: representative kymograph of dispersal data from a Δ*rbmA* replicate, representing the change in local density at 10-min intervals and at the indicated distances from the biofilm core. Black line represents the biofilm boundary. Right panel: as in the left panel for a model of random cell departures. Bottom panel: centroids of the density changes from the experimental data and the model of random cell departures over time. Lines and shading represent means and standard deviations for *N* = 5 biofilms, respectively. (D) Quantification of the radial components of the displacements during the dispersal phase for the indicated strains. Crossbars represent means and dots represent individual replicates for *N* = 5 biofilms, with the radial displacements averaged over each biofilm. *P* = 0.151 based on a two-sided unpaired *t* test relative to the wild-type data. (E) Quantification of dynamic region volume fraction during the dispersal phase for the indicated strains. Crossbars represent means and dots represent individual replicates for *N* = 5 biofilms. *P* = 0.067 based on a two-sided unpaired *t* test relative to the wild-type data. a.u., arbitrary units. n.s., *P* > 0.05. Underlying data for this figure can be found on Figshare (https://figshare.com/s/e0978ade2bc95dccf357).

deletion of a single gene involved in biofilm structural maintenance might perturb wild-type biofilm dispersal patterns in the reverse direction. To this end, we hypothesized that deletion of a gene encoding a biofilm matrix protein, *rbmA*, might enhance dispersal and therefore perturb the wild-type dispersal patterns in a manner opposed to the dispersal defect mutants. It has previously been shown that RbmA mediates cell–cell connections during biofilm

development and that deletion of *rbmA* does not prevent biofilm formation (Fig 5A) [13,37]. Rather, it reduces the local cell density in biofilms, particularly near the core. However, the relation of *rbmA* to dispersal has, to our knowledge, never been studied. When we quantified the change in biovolume over time for Δ*rbmA* biofilms, we found that dispersal was nearly complete (97% to 100% of the biofilm dispersed), unlike in the wild-type strain (Fig 5B). Strikingly, cell departure was initially slow, but approximately 7 h after the initiation of dispersal, a large fraction of each biofilm (in some cases >50% of the biofilm biovolume) rapidly dispersed in a 10- to 30-min window (Fig 5B and S6 Movie). When we then examined the spatial pattern of dispersal over time, we found that cell departures closely followed a random radial pattern (Figs 5C and S9), in contrast to the wild-type strain. These findings suggest that the maintenance of RbmA-mediated cell–cell connections is integral to the outside-in-to-random dispersal pattern and the existence of non-dispersing cells in the core of wild-type biofilms.

We next examined biofilm compression and dynamic region patterns as described above (Fig 3). We hypothesized based on our findings that these patterns are driven by local mechanical differences inside wild-type biofilms, which in turn, might emerge from local differences in the strength and number of cell–cell connections, as has been proposed previously [13,37,38]. Indeed, previous work has shown that the Δ*rbmA* mutant forms mechanically weak, fluid-like biofilms, with cells being loosely packed compared to the wild type [13,37,38]. Therefore, we hypothesized that the Δ*rbmA* mutant would lack the mechanical heterogeneity necessary for the development of dynamic regions and that the weakened cell–cell connections could attenuate compression. Indeed, we found that both patterns were, on average, reduced in the Δ*rbmA* strain compared to wild-type biofilms (Fig 5D and 5E and S6 Movie), though because of variation between replicates, the comparison did not attain statistical significance ($P > 0.05$). Taken together, we propose a model in which RbmA-mediated cell–cell linkages form a rigid scaffold that restricts local cell displacements. During dispersal, via an unknown mechanism, the number or strength of these linkages is diminished in certain regions of the biofilm, and due to differential pressure, the cells in these regions are pushed out, manifesting in visible dynamic regions. This remaining scaffold of cell–cell connections then pulls non-dispersing cells together to fill the empty space.

## Discussion

For decades, bacterial biofilm development has been conceived as a dynamic process that is patterned in space and time at the single-cell level according to elaborate cell-signaling and environmental inputs. Yet, the spatial and temporal progression of biofilm developmental transitions, especially biofilm dispersal, has been underexplored, in large part because the environmental sensitivity of fluorescent proteins makes biofilm cells difficult to investigate. Here, we leveraged FAPs to overcome these limitations and reveal the dynamics of biofilm dispersal in the global pathogen *V. cholerae* at high spatiotemporal resolution. We find that for wild-type *V. cholerae*, dispersal is a heterogeneous process, with approximately 25% of the cells in a biofilm remaining once the dispersal phase is complete, suggesting that cells can develop distinct fates. Going forward, we plan to use FAP reporters to examine the gene expression patterns that we presume underlie these distinct cell types. Moreover, future work will be required to determine whether the cells remaining after the dispersal phase exhibit physiological differences from their dispersing counterparts, whether they retain the benefits of the biofilm lifestyle (e.g., protection from threats), and whether they are able to re-seed new communities. We note that heterogeneous dispersal patterns are expected from previous theoretical work, even in spatially uniform environments, consistent with our observations [39]. This theoretical work suggests that heterogeneous cell dispersal from biofilms may be common

in bacteria—a hypothesis that we plan to test by applying the FAP labeling technology to other notorious biofilm formers.

We find that the timing of cell dispersal in *V. cholerae* biofilms initially depends on a cell's position relative to the biofilm core. Moreover, 2 dominant patterns emerged from our analysis: first, early in the dispersal phase, we observed the development of localized dynamic regions, where cells exhibited large outward displacements compared to the cells in surrounding regions, presumably functioning as dispersal "hot spots." The second dominant pattern we observed was that non-dispersing cells in the biofilm periphery compressed toward the biofilm core as internal cells dispersed. Although the location and magnitude of compression correlated with interior cell dispersal in the wild-type strain, interestingly, it was reduced even in the Δ*rbmA* strain, which disperses more than the wild-type strain. We speculate that this is a result of the differential mechanical properties of the wild-type strain, which is more solid-like, and the Δ*rbmA* strain, which is more fluid-like, as has been studied extensively previously [13,37,38]. Together, our findings may link previous work on the mechanical properties of *V. cholerae* biofilms with emergent patterns during a major lifestyle transition.

We anticipate that the FAP-fluorogen labeling approaches described here will enable research in diverse areas of microbiology. Specifically, the oxygen independence of FAP fluorescence combined with the far-red spectral properties of the dL5-MG complex could enable interrogation of microbiome biogeography in live animals in unprecedented detail. Moreover, distinct FAP-fluorogen pairs, beyond the dL5-MG complex used here, have been developed to span the visible spectrum [19,20]. These probes could be used to tag proteins of interest, to label multiple organisms simultaneously, or to monitor gene expression in complex polymicrobial communities. Thus, we propose that extensions of the FAP technology employed in this work could enable a mechanistic understanding of microbial community development from the level of single species to the level of complex multispecies assemblages.

## Methods

### Bacterial growth conditions, strains, and reagents

The parent *V. cholerae* strain used in this study was wild-type O1 El Tor biotype C6706str2. Antibiotics, when necessary, were used at the following concentrations: polymyxin B, 25 μg/ml; kanamycin, 50 μg/ml; spectinomycin, 200 μg/ml; gentamycin, 25 μg/ml. Strains were propagated in lysogeny broth (LB) supplemented with 1.5% agar or in liquid LB with shaking at 30°C. All strains used in this work are reported in S1 Table. For microscopy and bulk fluorescence measurements, *V. cholerae* cells were grown in M9 medium containing glucose and casamino acids (1× M9 salts, 100 μM $CaCl_2$, 2 mM $MgSO_4$, 0.5% dextrose, 0.5% casamino acids). *P. aeruginosa* and *E. coli* were grown in M63 minimal medium (1× M63 salts, 0.5% dextrose) and M9 minimal medium containing galactose and casamino acids (1× M9 salts, 100 μM $CaCl_2$, 2 mM $MgSO_4$, 0.5% galactose, 0.5% casamino acids), respectively.

Fluorogenic compounds were acquired via the Carnegie Mellon University Technology Transfer & Enterprise Creation office or as a gift of the Armitage Laboratory and the Molecular Biosensor and Imaging Center (MBIC) at Carnegie Mellon University. The fluorogens used in this study were malachite green ester, malachite green 2P, and MHN-ester (4-{[Bis (4-dimethylamino-phenyl)-methylene]-amino}-butyric acid ethyl ester). Fluorogens were dissolved and stored in ethanol with 1% acetic acid and were added to media, preceding inoculation, at 1 μM (for MGe, MG-2P) or 5 μM (for MHNe), unless otherwise stated. Solvent concentrations were maintained below 0.5% in culture media in all cases. To monitor the effects of FAPs and fluorogens on the growth of *V. cholerae*, $OD_{600}$ was monitored over time

at saturating fluorogen concentrations in 96-well plates using an Agilent Biotek Cytation 1 imaging plate reader driven by Biotek Gen5 (Version 3.12) software.

### DNA manipulation and strain construction

Modifications to the *V. cholerae* genome were generated by replacing genomic DNA with linear DNA introduced by natural transformation on chitin, as described previously [40,41]. PCR and Sanger sequencing (Azenta) were used to verify genetic alterations. *dL5* sequences were obtained from Szent-Gyorgyi and colleagues (2022) [42] and were codon optimized for expression in *V. cholerae*, and in all cases, FAP expression was driven by the tac promoter expressed from the neutral locus *vc1807*. Complete sequence information for the *mNeonGreen-dL5*, *SS-dL5*, and *dL5-µNS* constructs is available in S2 Table. Oligonucleotides and synthetic linear DNA g-blocks used in cloning were ordered from IDT and are reported in S3 Table. For expression in *E. coli* and *P. aeruginosa*, the *Ptac-mNeonGreen-dL5* construct was cloned into the broad host range vector pBBR1 using HiFi assembly (New England Biolabs).

### Anaerobic bacterial growth

The indicated *V. cholerae* strains were initially cultured in LB medium at 30°C overnight for 16 h with shaking and the following morning cultures were washed into M9 medium by centrifugation (13,000 rpm) and resuspension. Cultures were diluted 20-fold into 1.4 ml of M9 medium containing 300 nM MG-ester in air-sealed glass screw neck vials with PTFE/silicone Septum (Chem Glass Life Sciences). Following Hungate [43], for anaerobic conditions, atmospheric air was replaced by placing the culture under a stream of argon gas for 5 min. The argon was supplied through the septum with a needle and an additional needle was placed in the septum to remove air and excess argon. Cultures were grown anaerobically or aerobically overnight for 16 h and subsequently fixed by adding 4% formaldehyde in phosphate buffered saline. Bulk culture fluorescence was subsequently measured on an Agilent Biotek Cytation 1 imaging plate reader driven by Biotek Gen5 (Version 3.12) software. mNeonGreen signal was measured using a YFP filter set (ex. 500, em. 542) and dL5-MG-ester signal was measured using a Cy5 filter set (ex. 628, em. 685). Fluorescence output for each channel was divided by the measured $OD_{600}$ for each condition to normalize for cell numbers. A background subtraction was performed for each channel by subtracting the measured fluorescence of the wild-type strain, expressing neither mNeonGreen nor dL5. Values were subsequently internally normalized to the means of the oxygenated samples for ease of comparison.

### Microscopy

To image *V. cholerae* biofilms over time, starter cultures were first prepared by growing *V. cholerae* strains, inoculated from single colonies, in 200 µl of LB medium in 96-well plates covered with a breathe-easy membrane (Diversified Biotech, Dedham, Massachusetts, United States of America) to minimize evaporation. Cultures were grown overnight with constant shaking at 30°C. Saturated overnight cultures were subsequently diluted in 96-well plates for imaging to an $OD_{600}$ of $1 \times 10^{-5}$—roughly a 1:200,000 dilution in M9 medium containing glucose and casamino acids (1× M9 salts, 100 µM $CaCl_2$, 2 mM $MgSO_4$, 0.5% dextrose, 0.5% casamino acids). For brightfield and widefield approaches (Fig 1B) polystyrene 96-well microtiter dishes were used (Corning). For spinning disc confocal microscopy of biofilm dispersal, glass-bottom 96-well microtiter dishes (Mattek) were used. In all cases, the environmental temperature was maintained at 30°C throughout the duration of the imaging experiments.

To monitor the bulk fluorescence over biofilm growth (Fig 1B), *V. cholerae* cells were grown in the presence of 100 µM norspermidine (Millipore Sigma, I1006-100G-A) to enable

robust biofilm formation and inhibit dispersal. Single-plane images were captured using a 10× air objective (Olympus Plan Fluorite, NA 0.3) at 30-min intervals on an Agilent Biotek Cytation 1 imaging plate reader driven by Biotek Gen5 (Version 3.12) software. mNeonGreen excitation was achieved using a 505 nm LED and fluorescence was filtered with a YFP imaging cube (ex. 500, em. 542). dL5-MG-ester was excited using a 623 nm LED and was measured using a Cy5 imaging filter cube (ex. 628, em. 685).

Spinning disc confocal timelapse imaging was performed using a motorized Nikon Ti-2E stand outfitted with a CREST X-Light V3 spinning disk unit, a back-thinned sCMOS camera (Hamamatsu Orca Fusion BT) and a 100× silicone immersion objective (Nikon Plan Apochromat, NA 1.35) driven by Nikon Elements software (Version 5.42.02). The source of illumination was an LDI-7 Laser Diode Illuminator (89-North). Cells were labeled with 1 μM of the fluorogen malachite green coupled to diethylene glycol diamine (MG-2P), which fluoresces upon stable binding to the FAP dL5, which our strains produced. Unless otherwise stated, fluorogens were added at least 2 h prior to the initiation of the experiments. To achieve robust dL5 expression, strains harbored *Ptac-ssMBP-dL5* chromosomally integrated at the *vc1807* neutral locus. The *tac* promoter drove strong constitutive expression and the secretion signal of maltose-binding protein (ssMBP) was used to target dL5 to the periplasm. dL5-MG-2p fluorescence was excited at 640 nm. After initial dilution, as noted above, samples were grown statically at 30°C for 9 h and were subsequently transferred to a temperature-controlled chamber for microscopy (Oko Labs) at 30°C. Image acquisition was initiated 30 min later.

## Measurement of fluorogen-FAP EC$_{50}$ and labeling kinetics in biofilms

To determine half maximal effective concentrations (EC$_{50}$) of FAP-fluorogen complexes in *V. cholerae*, fluorogens were titrated at the indicated concentrations and added to M9 media in 96-well plates. The indicated strains were then inoculated into wells at an initial OD$_{600}$ of $10^{-4}$ and allowed to grow in the presence of their cognate fluorogens for 16 h. Endpoint bulk fluorescence readings were captured using a FITC filter set (ex: 485 nm, em: 523 nm) for MHNe or a Cy5 filter set (ex: 620 nm, em: 680 nm) for MGe and MG-2P. Raw fluorescence values were background subtracted by the fluorescence of the parent wild-type strain (no encoded FAP), grown in the presence of the appropriate fluorogen at the same concentration. Fluorescence values for each culture were then normalized to their OD$_{600}$ reading to correct for cell numbers. Normalized fluorescence values were subsequently plotted as a function of fluorogen concentration and fit with a Hill equation to determine the EC$_{50}$ (S3 Fig).

To determine how biofilm labeling kinetics differ for cytoplasmic (*mNeonGreen-dL5*) versus periplasm-targeted dL5 (*SS-dL5*), we first pre-grew biofilms of each strain in 96-well plates in the absence of fluorogens in M9 media with 100 μM norspermidine added to stimulate robust biofilm formation. After 20 h of growth, biofilms were washed 6× with 200 μl of fresh M9 media to remove planktonic cells. Samples were then placed on the microscope at 30°C and the focal position of biofilms in each well was defined using brightfield microscopy. Spinning disc confocal timelapse imaging was performed using a 20× air objective (Nikon Plan Apochromat, NA 0.75) and after 2 time points in the absence of fluorogens, saturating concentrations of each fluorogen were added and mixing was performed. Single-plane images, proximal to the coverslip, were captured at 2-min intervals. After the acquisition, biofilms were segmented in the brightfield channel and total far-red fluorescence was measured over time for each condition.

## Image analysis procedures

All image analysis procedures were performed using the Julia programming language (version 1.10.2) [44,45].

Image preprocessing: For all analyses of biofilm timelapses performed with high spatial and temporal resolution, we applied the same image processing steps. For each timelapse, we reduced the contribution of Poisson shot noise in the images using the Denoise.ai module available in Nikon Elements software (Version 5.42.02). We then performed top-hat filtering of each x-y slice to reduce the contribution of out-of-focus light along the optical axis. Next, to correct for x-y drift, we registered each timelapse to the central time point using the single-step discrete Fourier transform algorithm described by Guizar-Sicairos and colleagues [46]. We eliminated planktonic (non-biofilm-associated) cells from the timelapse using a custom algorithm that leverages the connectivity and continuity of biofilm cell positions over time. Specifically, reconstruction by dilation of consecutive binarized z-stacks (time points) was used to eliminate cells at time t that were not present or connected to biofilm cells at time t-1. Additionally, planktonic cells spontaneously appeared at high levels several hours after the beginning of the timelapse due to the initiation of dispersal. The specific time when this occurred was identified, and all voxels outside of the biofilm volume at that time point were omitted from the mask at subsequent time points. Finally, to further refine the planktonic cell removal, we identified the area in the x-y plane occupied by the biofilm based on a z-projection (as a proxy for how many layers of cells were present at each location). Cells lying outside this area in the x-y plane were removed. Together, these processes yielded a mask of biofilm cells at each time point, which was applied to the original timelapse to isolate biofilm cells.

Kymographs and random model testing: To compute kymographs (e.g., Fig 2C), images were down-sampled 4-fold and subsequently blurred with a Gaussian filter. For every time-lapse, we computed the distance transform relative to the biofilm core, which was determined using connected components analysis on a mask of the first image in the timelapse. We then iterated over binned distances from the biofilm core and computed the change in biofilm density (cellular volume fraction inside a bin) between consecutive time points in each bin using a bin side-length of approximately 2 μm. The result was rescaled to density change per hour to yield the kymograph representation. The boundary was computed as the maximum distance of a binarized voxel from the biofilm core, again using the distance transform. For the random model, we computed a cellular mask for every image in a timelapse, calculated the number of voxels that switched from a 1 to a 0 between that time point and the next from the data, and then randomly redistributed the departing voxels across the biofilm volume and computed the resulting density change inside the same bins as before. The result is a kymograph where, at every time point, the value of the density change per hour in a bin represents the outcome one would observe if voxels switched from 1 to 0 irrespective of their location with respect to the biofilm core. Centroids were extracted by computing the weighted average position of the density change data at every binned distance from the biofilm core and at every time point, where the weight for a given binned distance at time point t was its corresponding density change.

Calculating intra-biofilm displacements with image cross-correlation: We computed cellular displacements within each biofilm based on the confocal timelapses using cross-correlation methods (often called particle image velocimetry in the field of fluid mechanics) [47–50]. Briefly, each pair of consecutive z-stacks in a timelapse (with planktonic cells removed) was divided into interrogation windows overlapping by half their side lengths. The displacements between the 2 time points was evaluated on the resulting rectilinear grid; the cross-correlation between each interrogation window at time point t and time point t+1 was computed efficiently via the Fourier transform, owing to the convolution theorem. The peak of the cross-correlation tensor was used to extract the putative translation for a given interrogation window. Spurious displacements were detected by local median filtering, by thresholding the signal-to-noise ratio for each cross-correlation tensor, and by placing a global constraint on the length of each displacement vector. To maximize the spatial resolution of the displacement

field without generating additional spurious vectors, iterative window deformation was used with interrogation window side lengths of 64, 32, and 16 voxels (approximately 4, 2, and 1 μm) [49].

Biofilm compression and dynamic regions: Global biofilm compression was assessed by computing the radial component of each displacement vector with respect to the center of mass at the bottom of the largest connected component in the first image of the timelapse (taken to be the biofilm at the first time point). For the vector display in, for example, Fig 3B, the vectors were summed during the growth and dispersal phases, respectively, in a single x-z plane of interrogation windows in the middle of the biofilm. Spurious vectors were removed, and the radial components of the summed vectors were computed and displayed by color (see, for example, Fig 3B). To compute the average radial displacements (e.g., Fig 3C), we simply averaged the radial components for all vectors in the biofilm during the same growth and dispersal phases. Dynamic regions were defined by thresholding the offset between the radial displacement of a given vector and its local average during the 1.5 h preceding and 2.67 h following the onset of dispersal. The threshold was set as the local mean plus 260 nm, which achieved the greatest distinction between dynamic regions and the rest of the biofilm across all mutants and their respective replicates. The volume occupied by dynamic regions relative to the total biofilm volume was determined by summing the volumes of the interrogation windows (see above) that contained a vector with length greater than the threshold.

## Supporting information

**S1 Fig. Representative images of cells expressing cytoplasmic dL5, labeled with 1 μM MGe.**
(A) *E. coli*. (B) *P. aeruginosa* PA14. Scale bar is 10 μm.
(TIFF)

**S2 Fig. FAP-expressing strains grow normally in the presence of fluorogens.** (A) Growth curves at 30˚C for a strain constitutively expressing the cytoplasmic fluorescent protein mNeonGreen (*Ptac-mNeonGreen*) compared to strains constitutively expressing dL5 targeted to the cytoplasm (Ptac-mNeonGreen(Y69G)-dL5) or the periplasm (*Ptac-SS-dL5*), with no fluorogen added. (B) Growth curves for the indicated strains grown in the absence or presence of their cognate fluorogens at the indicated concentrations. In all cases, N = 3 biological replicates. Points represent means and shading represents standard deviations. Underlying data for this figure can be found on Figshare (https://figshare.com/s/e0978ade2bc95dccf357).
(TIFF)

**S3 Fig. FAP-fluorogen activation properties in *V. cholerae*.** (A) Whole-culture fluorescence intensity dose-response curves for MGe and MHNe, added to cells expressing *Ptac-mNeon-Green(Y69G)-dL5*, and MG-2P added to cells expressing *Ptac-SS-dL5*, measured after 16 h of growth in the presence of each fluorogen. Data were background subtracted and the dose-response relationship was fitted using a Hill equation, assuming no ligand binding cooperativity (Hill coefficient = 1). N = 3 biological replicates. Points represent averages, and error bars represent standard deviation. (B) Whole-culture fluorescence for the indicated conditions. Cells expressing cognate FAPs were grown for 16 h in the presence of MGe (1 μM), MHNe (5 μM), or MG-2P (1 μM). Background conditions were defined by growing wild-type V. cholerae (not expressing any FAP) in the presence of saturating fluorogen concentrations in M9 minimal medium. FC = fold change relative to background. N = 3 biological replicates. Points represent individual replicates and crossbars represent averages. (C) Images (left panel) and quantification of peak-normalized fluorescence intensity (right) as a function of time after addition of 1 μM MGe or 1 μM MG-2P to 20-h pre-grown biofilm communities expressing

*Ptac-mNeonGreen(Y69G)-dL5* or *Ptac-SS-dL5*, respectively. Images are contrasted separately for each FAP/fluorogen pair (as the periplasmic MG-2P labeling approach yields dramatically brighter biofilm communities). a.u., arbitrary units. Underlying data for this figure can be found on Figshare (https://figshare.com/s/e0978ade2bc95dccf357).
(TIFF)

**S4 Fig. Spatiotemporal biofilm dispersal patterns in individual wild-type replicates. Left-most column: kymographs of dispersal patterns, representing the change in local density at 10-min intervals and at the indicated distances from the biofilm core.** Black line represents the biofilm boundary. Each panel represents data from a single biofilm for N = 5 biofilms. Second column from the left: as in the left-most column for the "Random" model. The rate of overall cell departure was set to match the experimental data for each biofilm. Second column from the right: centroid position of the density change data in the left panels over time. Right-most column: spatial distribution of the local density profiles for the data and random model at the completion of dispersal. a.u., arbitrary units. Underlying data for this figure can be found on Figshare (https://figshare.com/s/e0978ade2bc95dccf357).
(TIFF)

**S5 Fig. Biofilm expansion and compression in individual wild-type replicates.** Each pair of "Growth" and "Dispersal" panels represents the displacement vectors during a ~1.5-h period preceding dispersal ("Growth") and throughout dispersal ("Dispersal") for an individual wild-type biofilm. Vector color represents the radial displacement with respect to the core of the biofilm. For ease of demonstration, a single slice through the y-axis is shown. Colorbar is the same for all panels. Underlying data for this figure can be found on Figshare (https://figshare.com/s/e0978ade2bc95dccf357).
(TIFF)

**S6 Fig. Spatiotemporal biofilm dispersal kymographs for individual Δ*cheY* replicates.** Left column: kymographs of dispersal patterns, representing the change in local density at 10-min intervals and at the indicated distances from the biofilm core. Black line represents the biofilm boundary. Each panel represents data from a single biofilm for N = 5 biofilms. Middle column: as in the left column for the "Random" model. The rate of overall cell departure was set to match the experimental data for each biofilm. Right column: centroid position of the density change data from the left and middle panels over time. Underlying data for this figure can be found on Figshare (https://figshare.com/s/e0978ade2bc95dccf357).
(TIFF)

**S7 Fig. Spatiotemporal biofilm dispersal kymographs for individual Δ*lapG* replicates.** Left column: kymographs of dispersal patterns, representing the change in local density at 10-min intervals and at the indicated distances from the biofilm core. Black line represents the biofilm boundary. Each panel represents data from a single biofilm for N = 5 biofilms. Middle column: as in the left column for the "Random" model. The rate of overall cell departure was set to match the experimental data for each biofilm. Right column: centroid position of the density change data from the left and middle panels over time. Underlying data for this figure can be found on Figshare (https://figshare.com/s/e0978ade2bc95dccf357).
(TIFF)

**S8 Fig. Spatiotemporal biofilm dispersal kymographs for individual Δ*rbmB* replicates.** Left column: kymographs of dispersal patterns, representing the change in local density at 10-min intervals and at the indicated distances from the biofilm core. Black line represents the biofilm boundary. Each panel represents data from a single biofilm for N = 5 biofilms. Middle column:

as in the left column for the "Random" model. The rate of overall cell departure was set to match the experimental data for each biofilm. Right column: centroid position of the density change data from the left and middle panels over time. Underlying data for this figure can be found on Figshare (https://figshare.com/s/e0978ade2bc95dccf357).
(TIFF)

**S9 Fig. Spatiotemporal biofilm dispersal kymographs for individual Δ*rbmA* replicates.** Left column: kymographs of dispersal patterns, representing the change in local density at 10-min intervals and at the indicated distances from the biofilm core. Black line represents the biofilm boundary. Each panel represents data from a single biofilm for N = 5 biofilms. Middle column: as in the left column for the "Random" model. The rate of overall cell departure was set to match the experimental data for each biofilm. Right column: centroid position of the density change data from the left and middle panels over time. Underlying data for this figure can be found on Figshare (https://figshare.com/s/e0978ade2bc95dccf357).
(TIFF)

**S1 Movie. Representative timelapse of WT *V. cholerae* biofilm dispersal.** Left panel: representative x-y slice. Right panel: x-z projection. Scale bar and time stamp as indicated.
(AVI)

**S2 Movie. x-y projection illustrating dynamic region formation of *V. cholerae* cells in biofilm dispersal.** Arrow points to representative dynamic region. Scale bar and time stamp as indicated.
(AVI)

**S3 Movie. Representative timelapse of Δ*cheY V. cholerae* biofilm dispersal.** Left panel: representative x-y slice. Right panel: x-z projection. Scale bar and time stamp as indicated.
(AVI)

**S4 Movie. Representative timelapse of Δ*lapG V. cholerae* biofilm dispersal.** Left panel: representative x-y slice. Right panel: x-z projection. Scale bar and time stamp as indicated.
(AVI)

**S5 Movie. Representative timelapse of Δ*rbmB V. cholerae* biofilm dispersal.** Left panel: representative x-y slice. Right panel: x-z projection. Scale bar and time stamp as indicated.
(AVI)

**S6 Movie. Representative timelapse of Δ*rbmA V. cholerae* biofilm dispersal.** Left panel: representative x-y slice. Right panel: x-z projection. Scale bar and time stamp as indicated.
(AVI)

**S1 Text. FAP labeling equilibria and kinetics.** Measurement of binding properties of FAPs expressed in V. cholerae.
(DOCX)

**S1 Table. Strains used in this study.** List of strains used in the study.
(DOCX)

**S2 Table. FAP-fusion sequences used in this study.** Color coded DNA sequences for FAP expression.
(DOCX)

**S3 Table. Oligos and synthetic DNA used in this study.** List of synthetic DNA used in the study.
(DOCX)

## Author Contributions

**Conceptualization:** Jojo A. Prentice, Robert van de Weerd, Andrew A. Bridges.

**Data curation:** Jojo A. Prentice, Sandhya Kasivisweswaran, Robert van de Weerd, Andrew A. Bridges.

**Formal analysis:** Jojo A. Prentice, Robert van de Weerd, Andrew A. Bridges.

**Funding acquisition:** Andrew A. Bridges.

**Investigation:** Jojo A. Prentice, Sandhya Kasivisweswaran, Robert van de Weerd, Andrew A. Bridges.

**Methodology:** Jojo A. Prentice, Robert van de Weerd, Andrew A. Bridges.

**Project administration:** Jojo A. Prentice, Andrew A. Bridges.

**Resources:** Jojo A. Prentice, Robert van de Weerd, Andrew A. Bridges.

**Software:** Jojo A. Prentice.

**Supervision:** Jojo A. Prentice, Andrew A. Bridges.

**Validation:** Jojo A. Prentice, Robert van de Weerd, Andrew A. Bridges.

**Visualization:** Jojo A. Prentice, Robert van de Weerd, Andrew A. Bridges.

**Writing – original draft:** Jojo A. Prentice, Andrew A. Bridges.

**Writing – review & editing:** Jojo A. Prentice, Sandhya Kasivisweswaran, Robert van de Weerd, Andrew A. Bridges.

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
