## [Editor Report · Decision Letter 0]

16 Aug 2024

Dear Dr Bridges, 

Thank you for submitting your manuscript entitled "Biofilm dispersal patterns revealed using far-red fluorogenic probes" for consideration as a Methods and Resources Article by PLOS Biology. Please accept my sincere apologies for the delay in getting back to you as we consulted with an academic editor about your submission. 

Your manuscript has now been evaluated by the PLOS Biology editorial staff, as well as by an academic editor with relevant expertise, and I am writing to let you know that we would like to send your submission out for external peer review.

Once your full submission is complete, your paper will undergo a series of checks in preparation for peer review. After your manuscript has passed the checks it will be sent out for review. To provide the metadata for your submission, please Login to Editorial Manager (https://www.editorialmanager.com/pbiology) within two working days, i.e. by Aug 18 2024 11:59PM.

Kind regards,

Richard

Richard Hodge, PhD

rhodge@plos.org

PLOS

---

## [Decision Letter · Decision Letter 1]

16 Oct 2024

Dear Drew,

Thank you for your continued patience while your manuscript "Biofilm dispersal patterns revealed using far-red fluorogenic probes" went through peer-review at PLOS Biology. Please accept my sincere apologies for the long delays that you have experienced during the peer-review process. Your manuscript has now been evaluated by the PLOS Biology editors, an Academic Editor with relevant expertise, and by two independent reviewers.

In light of the reviews, which you will find at the end of this email, we are pleased to offer you the opportunity to address the comments from the reviewers in a revision that we anticipate should not take you very long. In addition, please also make sure to address the following data and other policy-related requests that I have provided below (A-B):

(A) Please ensure that each of the relevant figure legends in your manuscript include information on where the underlying data can be found (i.e. your Figshare deposition), and ensure your supplemental data file/s has a legend.

(B) Per journal policy, if you have generated any custom code during the course of this investigation, please make it available without restrictions. Please ensure that the code is sufficiently well documented and reusable, and that your Data Statement in the Editorial Manager submission system accurately describes where your code can be found. 

**IMPORTANT - SUBMITTING YOUR REVISION**

*Resubmission Checklist*

*Published Peer Review*

*PLOS Data Policy*

*Blot and Gel Data Policy*

Sincerely,

Richard

Richard Hodge, PhD

rhodge@plos.org

REVIEWS:

Reviewer #1: This manuscript reports the development of a novel fluorescent reporter technique that consists of a small protein, dl5, and one of several different fluoregenic dyes that only fluoresce when bound to dl5. This reporter method is then used to track biofilm growth and dispersal over time for WT Vibrio cholerae and a number of deletion mutant derivatives. The work documents the spatial pattern of biofilm dispersal in V. cholerae with very good quantitative detail, including the contributions of native biofilm structure (rbmA deletion), motility/motion reversal (cheY) and matrix depolymersae secretion (lapG, rbmB). 

As a primarily methods development paper, this manuscript is very exciting and in my opinion should be seen by as many microscopy researchers as possible, as quickly as possible. There are at least two tremendous benefits of the dl5-MGe method described here: (1) it allows for clear imaging in the absence of oxygen, a major field-limiting problem with live-imaging conventional fluorescent proteins, (2) its excitation/emission maxima (636/664) should be compatible when co-imaging with the modern conventional far-red fluorescnet proteins (e.g. mKate2 - 588/633). When used appropriately, this means that people formerly limited to imaging 3-4 channels in one sample can now image 4-5. The range of applications for this method are immense, to say the least. 

The biological results are a little more modest/descriptive but still very good and add to the still-limited literature on the mechanistic details of biofilm dispersal. This is an important topic that is severrely under-studied at the cellular spatial scale. I don't have much to comment on with these sections of the paper, but one point that didn't quite land in my opinion was the result regarding fluid channels opening up in biofilm clusters during the dispersal process. I don't see what the authors are referring to regarding channels in any of the images in the paper - are they just talking about the opened spaces between cells in biofilms from which some cells have dispersed? Calling this space "channels" makes a connection to the idea of fluid transport channels in macroscopic biofilms that seems inappropriate and potentially confusing to people who don't work on these sytems. I would advise the authors to change the language here, but this is more of a semantic issue than a technical one. 

Reviewer #2 (Erin Gloag, signs review): This manuscript is submitted to PLOS Biology for consideration as a Methods and Resources article. The new methods and resources presented is the use of fluorogen-activating proteins as a labelling technology to analyse biofilm formation of V. cholerae. A strength of this manuscript is that authors clearly demonstrate the utility of this technology over conventional fluorescent proteins to image biofilms, but they also use this technology to analyse biofilm dispersal and the spatiotemporal kinetics and cell-cell interactions during this stage of biofilm development. This is a well-written and really interesting manuscript that I enjoyed reading. The data presentation is also unique and clearly presents complicated time-lapse data. My main comment would be to expand the discussion to include some limitations of the technology. My specific comments are below. Finally the time-lapse microscopy was a joy to watch.

Major comments

1. Can authors comment on using this technology to image polymicrobial biofilms? For example, can different bacteria be tagged with different FAP sequences that bind to unique fluorogens?

2. Can authors comment on using this technology in in vivo animal models. Or would this technology only me amendable in vitro? 

3. Methods. Include number of technical and biological replicates for each assay

4. Methods. What software was used to perform the imaging analysis. The imaging analysis and data presentation is a strength of the manuscript, and increased transparency in the methods would increase the utility for the biofilm field.

Minor comments

5. For clarity, I would suggest including the statistics on the graphs, rather than just in the figure legend alone as this gets lost. 

6. Include error in Figures 2E and F.

7. Figure 4B. It is difficult to discern between the blue (cheY) and green (rbmB) lines. I would suggest picking a more contrasting colour for one of them (pink, purple or red?)

---

## [Editor Report · Decision Letter 2]

5 Nov 2024

Dear Drew,

On behalf of my colleagues and the Academic Editor, Victor Sourjik, I am pleased to say that we can accept your manuscript for publication, provided you address any remaining formatting and reporting issues. These will be detailed in an email you should receive within 2-3 business days from our colleagues in the journal operations team; no action is required from you until then. Please note that we will not be able to formally accept your manuscript and schedule it for publication until you have completed any requested changes.

PRESS

Best wishes, 

Richard

Richard Hodge, PhD

rhodge@plos.org

PLOS
